# Rehabilitation interventions delivered via telehealth to support self-management of rheumatic and musculoskeletal diseases: A scoping review protocol

**Rosemarie Barnett**[1], **Nuzhat Shakaib**[1], **Thomas A. Ingram**[1], **Simon Jones**[2], **Raj Sengupta**[3,4], **Peter C. Rouse**[1] *

1 Department for Health, University of Bath, Bath, United Kingdom, 2 Department of Computer Science, University of Bath, Bath, United Kingdom, 3 Royal National Hospital for Rheumatic Diseases, Royal United Hospitals Bath NHS Foundation Trust, Bath, United Kingdom, 4 Department of Pharmacy and Pharmacology, University of Bath, Bath, United Kingdom

* pr222@bath.ac.uk

## Abstract

### Background

Telerehabilitation is a term to describe rehabilitation services delivered via information and communication technology. Such services are an increasingly important component for the management of rheumatic and musculoskeletal diseases (RMDs). Telerehabilitation has the potential to expand the long-term self-management options for individuals with RMDs, improve symptoms, and relieve pressures on health care services. Yet, little is known about the variety of interventions implemented, and how they are being evaluated. Thus, this scoping review aims to identify and describe existing rehabilitation interventions delivered via telehealth for RMDs. Specifically, we aim to identify and summarize the key components of rehabilitation, the technology used, the level of health care professional interaction, and how the effectiveness of interventions is evaluated.

### Methods

We will conduct this review following the latest JBI scoping review methodology and the PRISMA guidelines for Scoping Reviews (PRISMA-ScR). The 'Population-Concept-Context (PCC)' framework will be used, whereby the 'Population' is RMDs ($\geq$18 years); the 'Concept' is rehabilitation; and the 'Context' is telehealth. Developed in collaboration with a subject Librarian, refined PCC key terms will be utilized to search (from 2011–2021) three electronic databases (i.e., Embase, Scopus, Web of Science) for articles published in English. Search results will be exported to the citation management software (EndNote), duplicates removed, and eligibility criteria applied to title/abstract and full-text review. Relevant information pertaining to the PCC framework will be extracted. Data will be summarized qualitatively, and if appropriate, quantitatively via frequency counts of the components comprising the 'Concept' and 'Context' categories of the PCC framework.

**Data Availability Statement:** No datasets were generated or analyzed during the current study. All

relevant data from this study will be made available upon study completion.

**Funding:** This work was supported by the Sir Halley Stewart Trust [grant number:2316, awarded to SJ, PR, RB; https://www.sirhalleystewart.org.uk/], who provided funding for the time of co-authors NS and TI. The funders did not and will not have a role in study design, data collection and analysis, decision to publish, or preparation of the manuscript.

**Competing interests:** I have read the journal's policy and the authors of this manuscript have the following competing interests: RS has received speaker fees, consultancy and/or grants from Abbvie, Biogen, Celgene, Lilly, MSD, Novartis, Roche and UCB. RS has represented Abbvie and Novartis at NICE technology appraisals. RB, NS, TI, SJ and PR have declared no competing interests. This does not alter our adherence to PLOS ONE policies on sharing data and materials.

## Discussion

Findings from the proposed scoping review will identify how telehealth is currently used in the delivery of rehabilitation interventions for RMDs. The findings will develop our understanding of such interventions and provide a platform from which to inform future research directions.

## Introduction

### Background and rationale

Rheumatic and musculoskeletal diseases (RMDs) are a complex, diverse group of chronic conditions often associated with immune dysfunction, inflammation, and gradual deterioration of joints, muscles and bones [1]. Many of these conditions result in significant pain and disability, which can greatly impact people's quality of life [1]. Effective management of RMDs includes multidisciplinary rehabilitation, aimed at enabling patients to self-manage behaviour, symptoms, and treatment, to reach and maintain their optimal physical, psychological, and social functioning [2–4]. Telehealth, and specifically telerehabilitation, has become a critical feature of RMD health care, particularly over the last 36 months as a result of the COVID-19 pandemic, to provide continued care to patients [5]. Yet, the recent enforced, and often rapid transition to remote delivery of care, has meant that little is known about the variety of interventions adopted and whether they are being evaluated. To help guide the future development, refinement, evaluation and effectiveness of such novel interventions, this scoping review aims to systematically search the recent literature to identify and describe existing rehabilitation interventions delivered via telehealth for RMDs and identify how their effectiveness is being evaluated. We will map the available literature/evidence base, identify knowledge gaps and clarify key concepts [6].

The World Health Organisation defines rehabilitation as "a set of interventions designed to optimize functioning and reduce disability in individuals with health conditions in interaction with their environment" [7]. Telehealth is the umbrella term used to describe the provision of health care at a distance using information and communication technology [8]. Thereby, telerehabilitation is the delivery of rehabilitation services via information and communication technology [9]. Many advantages to telerehabilitation have been proposed, such as allowing greater accessibility for patients who find it difficult to make it to clinic [10], maintaining continuity of care for outpatients [11], and reducing burden on health system resources [12]. Moreover, telerehabilitation allows patients to continue to self-manage their condition when access to usual care is restricted, such as during the COVID-19 pandemic. Yet, despite the advantages, pitfalls and limitations also need to be considered—with digital exclusion a prominent and pressing issue [5, 13–15]. The target population for telerehabilitation therefore needs to be carefully considered; with safety-nets in place to prevent widening of existing health inequalities [5].

The reduced peer support that is available to patients through remote delivery has also been highlighted as a potential issue in RMDs [16]. Similarly, the importance of in-person healthcare professional (HCP) interaction for reassurance has been recently highlighted; both for patients that they have been assessed holistically, and for staff, that they haven't missed key signs of disease progression/ deterioration [5]. The support provided by a HCP in telerehabilitation interventions can range from providing education tailored to the condition (e.g.,

lifestyle and diet advice, flare management) to creating a personalised physical activity action plan in conjunction with the patient, and monitoring progress remotely whilst providing feedback. Telerehabilitation interventions can also involve a varied approach to interacting with the patient, including virtual consultations by videoconference [17], the use of mobile phone applications (apps) to capture data in between consultations [18], and even blended care which combines face-to-face care with the use of telehealth [19]. The method of providing content and communicating with patients within these interventions can be performed synchronously (uses real-time technology to exchange information instantaneously between all users) or asynchronously (commonly referred to as 'store-and-forward', where there is a temporal delay between the sending and viewing of health information), or both. Thus, we anticipate heterogeneity in the design and implementation of telerehabilitation interventions to support self-management of RMDs. In particular, the level of support provided by HCPs within telerehabilitation interventions should be further explored, as this may play a role in adherence. Gamification has been utilised in the process of rehabilitation to increase adherence by implementing health-enabling technologies, whereby patients perform rehabilitation exercises through the use of play in gaming concepts [20].

While there is some evidence to suggest that telerehabilitation may be effective in improving symptoms in RMDs, evidence is still limited. Existing reviews have focused on musculoskeletal conditions more broadly including chronic non-malignant musculoskeletal pain [21], osteoarthritis, low-back pain, acute rehabilitation after surgery [22, 23], or rheumatoid arthritis alone [24]–which may not be relevant to other inflammatory RMDs such as axial spondyloarthritis (axSpA). In musculoskeletal conditions, systematic and meta-analytic reviews have found that real-time telerehabilitation (including acute rehabilitation post-surgery) can improve physical function and pain [23], and a randomised controlled trial (RCT) has shown that remote support via an app compared to paper handouts increases adherence to home-exercise programmes [25]. Good overall adherence to telerehabilitation, and improvement in symptoms, has been reported in knee osteoarthritis following a feasibility study using pre- and post-test design [10], and improvement in symptoms were observed for fibromyalgia following video-guided aerobic exercise as part of a RCT [26]. These positive findings suggest that telerehabilitation may also be effective in improving symptoms in inflammatory RMDs, via the provision of healthcare from a distance.

Telerehabilitation interventions are expected to provide an effective long-term solution to broaden the provision of self-management for RMDs and improve symptoms, whilst relieving pressure on health care systems [5, 10, 25–27]. However, neither the most effective transition methods, nor the long-term effects of the transition from usual care to telerehabilitation are known [16]. With the rapid uptake of telehealth delivered services by organisations following COVID-19, it seems plausible that the necessary rigorous and systematic steps required to ensure the success and sustainability of the telehealth model may have been overlooked; for example, how to maintain patient satisfaction and minimise patient burden, as well as the ability of HCPs to effectively deliver the intervention [28]. Overall, evidence on the efficacy and effectiveness of telerehabilitation in RMDs is limited, thus there is a need for an up-to-date, systematic review of the literature on telerehabilitation interventions in RMDs, and to provide evidence on their content, and how efficacy and effectiveness are being evaluated [22].

A scoping review will therefore be conducted to identify and describe existing, novel rehabilitation interventions delivered via telehealth in RMDs. A scoping review is deemed the best approach due to the broad, exploratory nature of this review [29, 30]. The 'PCC' framework will be utilised (Population, Concept and Context), whereby the Population is RMDs; the Concept includes the mechanisms and components of rehabilitation; and the Context (setting) is telehealth, specifically focusing on outpatient and home settings. We will also explore how the

effectiveness of the interventions are evaluated. The use of terminology in the literature is varied therefore the broader term of telehealth will be applied. The various mechanisms of rehabilitation utilised (e.g., behaviour change techniques, education, physical activity guidance, progress monitoring, psychological support) will also be summarised. We will explore interventions delivered via telehealth over the past 10 years as this period of time covers the gradual increase in the use of telehealth, including those studies prior to the COVID-19 pandemic.

## Aims and research questions of the scoping review

The primary aim of the proposed scoping review will be to identify and describe the existing rehabilitation interventions delivered via telehealth for RMDs and identify how effectiveness is evaluated.

Research questions of the scoping review:

- What aspects/components of rehabilitation (e.g., education, disease management, psychological support, physical activity, physiotherapy, behaviour change techniques) are delivered via telehealth for RMDs?

- What technology is used to deliver rehabilitation interventions for RMDs via telehealth?

- What type and level of HCP interaction exists in rehabilitation interventions delivered via telehealth for RMDs?

- How has effectiveness of rehabilitation interventions delivered via telehealth for RMDs been evaluated?

## Method

The scoping review will be performed following guidance from the JBI Manual for Evidence Synthesis on Scoping Reviews [29, 30]. Results will be reported in line with the PRISMA-ScR guidance (Preferred Reporting Items for Systematic reviews and Meta-Analyses extension for Scoping Reviews) [31]. The scoping review protocol has been registered on Figshare [32]. To establish the quality of our protocol, we have completed the PRISMA-P checklist, S1 Appendix. If required, amendments to the protocol will be documented (including what was changed, rationale) and reported in the final scoping review results publication/report.

## Eligibility criteria

**Inclusion.**  *Population.*

- Rheumatic or musculoskeletal disease diagnosis

- Aged ≥18 years

- Long-term condition which can be considered for self-management

  *Concept: Rehabilitation.*

- Intervention includes one or more aspect of rehabilitation: education, disease management, occupational therapy, psychological support, physical activity, physiotherapy, behaviour change techniques

- Intervention includes interaction with HCP during or at the end of the study

  *Context: Telehealth.*

- Intervention is delivered via telehealth

- Intervention is used for outpatient or home-based setting

**Exclusion.** *Population.*

- Studies in which patients are unable to make decisions (e.g. mentally incapacitated)

- Studies in which patients are immobile

- Diagnosis of non-specific pain disorders where pain is often the only symptom (e.g. chronic lower back pain, myofascial pain syndrome)

*Concept: Rehabilitation.*

- Intervention is associated with surgery, i.e. pre- or post-surgery rehabilitation

- Intervention developed for management of osteoporosis only (a common hallmark of RMDs)

- Intervention is used for prevention of a condition

- Intervention is used for inpatient settings

- Intervention is used for diagnosing and not managing a condition

*Context: Telehealth.*

- Any telehealth intervention with no HCP interaction (use of apps, sensors, wearables, gamification only)

## Types of evidence sources

- Evidence sources/publication types that will be included from the databases searched are detailed below:

  - Peer-reviewed articles,

  - Clinical trials (randomised / controlled),

  - Letters, editorials, notes, short survey,

  - Reference list scanning of included full-text articles will also be used.

- Types of evidence sources that will be excluded are provided below with reasons:

  - Conference review- this is a list of conference themes / topics which does not provide detailed information on intervention/ rehabilitation

  - Grey literature- research into telerehabilitation appears to be well documented in various publication types therefore it is not deemed necessary to review any grey literature

  - Abstracts—sufficiently detailed information on the intervention/ rehabilitation cannot be identified from the abstract alone

## Search strategy

In order to build the search strategy, two broad searches in PubMed were initially conducted to explore the literature and understand what type of terminology is used in this field. For

these initial broad searches, the following search terms were used for the first search: '(virtual rehabilitation) AND (rheumatic disease OR musculoskeletal disease)', followed by a second search using the terms: '(virtual rehabilitation) AND (disease management) AND (rheumatic disease OR musculoskeletal disease)'. From searching the foremost 80 titles and abstracts of the first search sorted by 'relevance', and the foremost 120 titles and abstracts of the second search also sorted by 'relevance', 13 relevant articles were identified, five of which had published search strategies [20, 33–36]. Using these search strategies, relevant search terms were extracted and categorised under the PCC headings, namely: Rheumatic and musculoskeletal disease, Rehabilitation, and Telehealth. The 'Population' terms were largely taken from a prior systematic literature review on mobile health apps in RMDs [37]. The core study team (NS, PR, RB, SJ) agreed on terms to be included in the final search by 3 or more votes for each search term. The PCC search terms were refined and updated based on input from an experienced subject librarian (PB), with clinical oversight from a Consultant Rheumatologist (RS) on the 'population' terms. The final search terms using the PCC framework are shown in Table 1. The literature search will be conducted in the following databases: Embase, Scopus, and Web of Science.

The database searching will be conducted by one researcher (NS) supported by a subject librarian (PB) using truncation, wildcards and proximity searching. Articles in English language only will be included to facilitate the abstract review. The publication dates 2011-9th Nov 2021 will be included in the scoping review as telehealth is still emerging, thereby studies prior to this date may lack relevant interventions. After the search has been run, a second step to build the search strategy will be to update the provisional search terms by reviewing the keywords/ indexed terms of the top 50 articles in each database by date (most recent), relevance and highest citations. Once the searches have been run in all databases, a third step will be to identify any relevant references cited in full-text articles that have not already been included. Search terms will then be updated to capture relevant missing articles. Search terms will be updated and refined until unanimous agreement is reached that all relevant articles have been identified, while maintaining a search focussed enough to not capture large quantities of irrelevant articles.

## Source of evidence selection

The references from the three database searches will be amalgamated into Endnote, and duplicates removed, before exporting into Excel. A small sample (~1%) of titles/abstracts will be selected for the full team (PR, NS, RB) to apply the eligibility criteria and check for agreement at full-text inclusion. This task serves as a pilot test of the eligibility criteria, to refine the criteria and ensure that key information is not missed and that the process works as intended. At this stage, the eligibility criteria for title/abstract screening can be amended and finalised based on any discrepancies or conflicts. Once the criteria are finalised, the remaining titles/abstracts will be screened by two independent reviewers (NS, RB).

**Title/abstract screening.** The first stage of evidence selection will be to review titles and abstracts for eligibility criteria in Excel. A 'Yes' or 'No' vote will be cast by each independent reviewer to determine inclusion of the article for full-text review, with reasons for exclusion coded based on the eligibility criteria. References that do not provide an abstract, or it is unclear from the title and abstract whether the article is relevant or not, will be voted as 'Yes' and selected for further full-text review. After all titles/abstracts have been reviewed, results will be compared and any discrepancies discussed and resolved, with the oversight of a third reviewer (PR).

**Table 1. Search terms using the PCC framework.**

| PCC Framework | Search termsS |
|---|---|
| **Population**: rheumatic and musculoskeletal diseases AND | musculoskeletal OR "connective tissue" OR Myositis OR *leroderma OR "Systemic Lupus Erythemato*" OR "s.l.e." OR "Sle" OR Sjogren* OR Sjoegren* OR Vasculitis OR Rheuma* OR "*inflammatory arthr*" OR arthritis OR "osteo-arthr*" OR Osteoarthr* OR "degenerative arthr*" OR "degenerative joint disease" OR "psoriatic arthr*" OR Spondylarthr* OR Spondyloarthr* OR "ankylosi* sp*" OR "axial sp*" OR "*htere* disease" OR "Spin* ankylosi*" OR "Spondyl* ankylo*" OR "Rheumatic and musculoskeletal dis*" OR RMD* OR Gout* OR "arthritis urica" OR "urate inflammation" OR "uric arthritis" OR Fibromyalgia OR Hypermobility |
| **Concept**: rehabilitation AND | Rehab* OR "Physical therapy" OR Exercis* OR fitness OR workout OR "work-out" OR "physical activity" OR Physiotherap* OR "Physical treatment" OR "Physio therap*" OR "Self manag*" OR "Disease manag*" OR "pain manag*" OR "medical manag*" OR "emotional manag*" OR "social manag*" OR "practical manag*" OR "physical manag*" OR "rehab* manag*" OR "psychological manag*" OR "psychosocial manag*" OR "disorder* manag*" OR "illness manag*" OR "manag* of disease*" OR "manag* of disorder*" OR "Patient activation" OR "patient education" OR "patient medication knowledge" OR "training progress" OR "training plan*" OR "therapeutic intervention" OR "Psychological treatment" OR Kinesiotherap* OR Kinesitherap* OR "Education program*" OR "Occupational therap*" OR "Home based training" OR "patient monitor*" |
| **Context**: virtual/remote contexts of rehabilitation (specifically focusing on outpatient and home setting) | Telecare OR Tele-care OR telehealthcare OR "tele-healthcare" OR "tele-health care" OR "mobile care" OR "Mobile medicine" OR "mobile health" OR "Phone care" OR "Phone medicine" OR "Phone health" OR "online care" OR "Online medicine" OR "Online health" OR "internet care" OR "internet medicine" OR "internet health" OR "digital care" OR "digital medicine" OR "digital health" OR "contactless care" OR "contactless medicine" OR "contactless health" OR telehealth OR "e-health" OR ehealth OR "tele-health" OR Telerehab* OR "e-rehab*" OR "remote rehab*" OR "tele-rehab*" OR "virtual rehab*" OR "Real-time rehab*" OR "Real-time consult*" OR "Real-time communica*" OR "Real-time treat*" OR "Real-time therap*" OR "Real-time assess*" OR Telemedicine OR "Tele-medicine" OR Telemonitor* OR "Tele-monitor*" OR "distant monitor*" OR "remote monitor*" OR telerheumatology OR "Tele-rheumatology" OR teletreatment OR "Tele-treatment" OR Teleconsult*OR "tele-consult*" OR "long distance consult*" OR "telephone consult*" OR "web-based intervention" OR "internet-based intervention" OR "internet-intervention" OR "online-based intervention" OR "online-intervention" OR "web intervention" OR mhealth OR "m-health" OR "Remote consult*" OR "Remote communica*" OR "Remote treat*" OR "Remote therap*" OR "Remote assess*" OR "Virtual consult*" OR "Virtual communica*" OR "Virtual treat*" OR "Virtual therap*" OR "Virtual assess*" OR "e-consult*" OR "e-communica*" OR "e-treat*" OR "e-therap*" OR "e-assess*" OR videoconferenc* OR "Video conferenc*" OR "Synchronous consult*" OR "Synchronous communica*" OR "Synchronous treat*" OR "Synchronous therap*" OR "Synchronous assess*" OR "Synchronous videoconferenc*" OR "Asynchronous consult*" OR "Asynchronous communica*" OR "Asynchronous treat*" OR "Asynchronous therap*" OR "Asynchronous assess*" OR "In-home tele*" OR "telecommunication-based" OR "outpatient rehab*" OR "Self-rehab*" OR "technology-assisted therap*" OR "computer-assisted rehab*" OR "Computer-based intervention" OR "computer-assisted therap*" OR "technology-based intervention" OR "technology-assisted rehab*" OR gamification OR exergam* OR "communication technology" OR app OR "Store-and-forward" OR Videophon* OR "Video-phon*" OR smartphon* OR "smart phon*" OR (digital NEAR/4 intervention*) OR "Blended care" OR Teletherapy OR "Tele-therapy" OR "Mobile application" |

**Full-text review.**   Once all titles/abstracts have been screened, full texts will be reviewed for inclusion. As for the abstract screening, a 'Yes' or 'No' vote will be cast in Excel by each reviewer (NS, RB), with reasons for exclusion coded based on the eligibility criteria. Again, modifications may be made to the criteria at this stage if deemed necessary by the two reviewers, through discussion. The remaining references will again be reviewed with the finalised criteria, if applicable, by two independent reviewers (NS, RB), with any discrepancies discussed and resolved with the oversight of a third reviewer (PR).

If there is no link to the full text, a copy will be requested from the Inter-Library Loans Service. If it is still not available, the authors will be contacted for a copy of the article. If after retrieving full-text articles the relevance of an article cannot be decided, the authors will be contacted to gain further clarification.

## Data extraction

Data Extraction will involve extracting details from relevant full text articles (those voted 'Yes' in the Full Text Review stage). This is known as data charting with details shown in S2 Appendix. The data charting table will also be piloted on two or three sources to ensure all relevant results are extracted. This pilot step will be conducted by two members of the study team (NS and RB). The remaining data will be extracted by three members of the study team (NS, RB, and TI).

## Updated search

An updated search of the literature will be conducted by one researcher (TI) prior to the final summarizing of results. Eligible articles identified since the date of the initial search (9th Nov 2021) will be included. The same title/abstract and full-text screening process as previously eluded, will be conducted by two independent reviewers (RB, TI), with conflicts resolved by of a third reviewer (PR). Any additional data will be extracted by two members of the research team (RB, TI).

## Summarizing and presenting the findings

The review process will be illustrated and summarised via the PRISMA 2020 flow diagram. Data will be extracted into the data charting table (S2 Appendix). This data will be summarized and include descriptive data (e.g., frequency counts of population characteristics such as gender and condition); mean age and mean duration of symptoms/time since diagnosis; and explicit details on type of study, intervention purpose, description/details of intervention, HCP interaction details, aspect of rehabilitation targeted, context of telehealth targeted, and how effectiveness is measured. A narrative overview of the aggregated findings will be presented, and the methodological quality of studies will not be assessed.

In addition to a narrative summary of information pertaining to the PCC framework, the components under the 'Concept' and 'Context' of the PCC framework will be summarized as frequency counts in tabular form if appropriate, (i.e., if the information in the articles can be confined in this way; see S3 Appendix). The references will be listed in each row and two columns will represent the 'Concept' and 'Context'. An 'x' will be placed in the table if the categories are examined in the study. All data provided as frequency counts will be presented in tables, and pictorially where appropriate such as pie charts/ bar charts.

## Discussion, strengths, and limitations

In recent years, telehealth has emerged as a feasible option to provide rehabilitation services for individuals with a range of RMDs. Telehealth interventions may be particularly important

in reducing the burden on health services, in addition to providing important recovery options for those unable able to travel due to disability or people living in rural areas. A recent scoping review has found that patient satisfaction and the effectiveness of synchronous telehealth was comparable to conventional approaches to rehabilitation for the management of musculoskeletal disorders [38]. Further, in a scoping review of articles covering a range of conditions (e.g., musculoskeletal, neurological, pulmonary, and cardiac), remotely delivered physiotherapy was found to be acceptable, feasible, and safe, with comparable effectiveness and lower in cost to in-person physiotherapy [39]. Yet, little is known about the content of telerehabilitation interventions in RMDs, the technology used, the type and level of HCP interaction, or how the effectiveness of such interventions are assessed. Therefore, the findings from the proposed scoping review will offer direction for future research, by identifying gaps and key concepts, and supporting the refinement and development of telerehabilitation interventions in RMDs.

In light of the aforementioned implications of conducting this work, it is important to consider the strengths and limitations. The findings will help to inform healthcare professionals working across a wide range of rheumatic and musculoskeletal conditions. Further, the search strategy was developed via a rigorous and iterative process to maximize relevant articles and reduce the number of missed articles. The scoping review spans over at least a 12-year period, and therefore the summarized findings include the general increase in telehealth research and articles arising in response to the COVID-19 pandemic. Multiple researchers will be involved in each stage of the review process, thus reducing the risk of bias. However, the main limitation of the proposed scoping review pertains to the sources of evidence. Despite telerehabilitation appearing to be well documented across a range of publication types, it is possible that some articles may be missed as the grey literature and articles published in a language other than English, will not be reviewed. Limitations will be further discussed in the proposed scoping review, in addition to any changes to the protocol.

## Supporting information

**S1 Appendix. PRISMA-P checklist.**
(DOCX)

**S2 Appendix. Data charting table.**
(DOCX)

**S3 Appendix. Charting evidence of 'Concept' and 'Context' for included studies.**
(DOCX)

## Acknowledgments

We acknowledge Subject Librarian for Health and Social & Policy Sciences Peter Bradley (PB) for their support in defining the study search terms.

## Author Contributions

**Conceptualization:** Rosemarie Barnett, Nuzhat Shakaib, Thomas A. Ingram, Simon Jones, Raj Sengupta, Peter C. Rouse.

**Data curation:** Rosemarie Barnett, Nuzhat Shakaib, Thomas A. Ingram, Peter C. Rouse.

**Funding acquisition:** Rosemarie Barnett, Simon Jones, Raj Sengupta, Peter C. Rouse.

**Investigation:** Rosemarie Barnett, Nuzhat Shakaib, Thomas A. Ingram, Peter C. Rouse.

**Methodology:** Rosemarie Barnett, Nuzhat Shakaib, Thomas A. Ingram, Simon Jones, Raj Sengupta, Peter C. Rouse.

**Project administration:** Rosemarie Barnett, Nuzhat Shakaib, Thomas A. Ingram, Peter C. Rouse.

**Resources:** Rosemarie Barnett, Nuzhat Shakaib, Thomas A. Ingram, Peter C. Rouse.

**Supervision:** Simon Jones, Raj Sengupta, Peter C. Rouse.

**Visualization:** Rosemarie Barnett, Nuzhat Shakaib, Thomas A. Ingram, Peter C. Rouse.

**Writing – original draft:** Rosemarie Barnett, Nuzhat Shakaib, Thomas A. Ingram, Simon Jones, Raj Sengupta, Peter C. Rouse.

**Writing – review & editing:** Rosemarie Barnett, Nuzhat Shakaib, Thomas A. Ingram, Simon Jones, Raj Sengupta, Peter C. Rouse.

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
