## [Decision Letter · Decision Letter 0]

10 Jan 2024

PONE-D-23-09456Rehabilitation interventions delivered via telehealth to support self-management of rheumatic and musculoskeletal diseases: A scoping review protocolPLOS ONE

Dear Dr. 

Thank you for submitting your manuscript to PLOS ONE. After careful consideration, we feel that it has merit but does not fully meet PLOS ONE’s publication criteria as it currently stands. Therefore, we invite you to submit a revised version of the manuscript that addresses the points raised during the review process.

We look forward to receiving your revised manuscript.

Kind regards,

Mehrnaz Kajbafvala, Ph.D

Academic Editor

PLOS ONE

Journal Requirements:

"I have read the journal's policy and the authors of this manuscript have the following competing interests: RS has received speaker fees, consultancy and/or grants from Abbvie, Biogen, Celgene, Lilly, MSD, Novartis, Roche and UCB. RS has represented Abbvie and Novartis at NICE technology appraisals. RB, NS, TI, SJ and PR have declared no competing interests."

Reviewers' comments:

Reviewer's Responses to Questions

**Comments to the Author**

1. Does the manuscript provide a valid rationale for the proposed study, with clearly identified and justified research questions?

Reviewer #1: Yes

2. Is the protocol technically sound and planned in a manner that will lead to a meaningful outcome and allow testing the stated hypotheses?

Reviewer #1: Yes

3. Is the methodology feasible and described in sufficient detail to allow the work to be replicable?

Reviewer #1: Yes

4. Have the authors described where all data underlying the findings will be made available when the study is complete?

Reviewer #1: Yes

5. Is the manuscript presented in an intelligible fashion and written in standard English?

Reviewer #1: Yes

6. Review Comments to the Author

You may also provide optional suggestions and comments to authors that they might find helpful in planning their study.

Reviewer #1: Rehabilitation interventions delivered via telehealth to support self-management of rheumatic and musculoskeletal diseases: A scoping review protocol

Thank you for the opportunity to review this manuscript. The authors have conducted: A scoping review protocol to identify interventions delivered via telehealth for rheumatic and musculoskeletal diseases. I find the manuscript interesting and within the scoop of PLOS ONE journal. Please find my comments below.

• I found a study similar with your review protocol! Please clarify this issue. (10.1093/rheumatology/keac133.287- Protocol for a scoping review to identify self-management support interventions delivered via telehealth for rheumatic and musculoskeletal diseases. Nuzhat Shakaib, Rosemarie Barnett, Raj Sengupta, Simon Jones, Peter C Rouse)

- Add registration number in method section.

- State plan for documenting important protocol amendments.

- I think it is better to add the PRISMA 2020 flow diagram for reporting screening result.

- What Quality appraisal tool will be used to evaluate the quality of the eligible studies.

7. PLOS authors have the option to publish the peer review history of their article (what does this mean?). If published, this will include your full peer review and any attached files.

Reviewer #1: **Yes: **Dr. Saeideh Babazadeh-Zavieh

---

## [Author Response · Author response to Decision Letter 0]

23 Feb 2024

Reviewer comment: Thank you for the opportunity to review this manuscript. The authors have conducted: A scoping review protocol to identify interventions delivered via telehealth for rheumatic and musculoskeletal diseases. I find the manuscript interesting and within the scoop of PLOS ONE journal. Please find my comments below. 

Author response: Thank you for taking the time to review our manuscript, and for your helpful comments below. We have responded to each one in detail and have revised the manuscript accordingly. 

Reviewer comment: I found a study similar with your review protocol! Please clarify this issue. (10.1093/rheumatology/keac133.287- Protocol for a scoping review to identify self-management support interventions delivered via telehealth for rheumatic and musculoskeletal diseases. Nuzhat Shakaib, Rosemarie Barnett, Raj Sengupta, Simon Jones, Peter C Rouse) 

Author response: Thank you – this is indeed an abstract version of the protocol, which was presented at BSR 2022. However, there is not enough detail provided in the abstract to ensure transparency of our methodology and to limit the occurrence of reporting bias (see section 11.2 in the JBI Manual for Evidence Synthesis). Thus, we have decided to also publish the full protocol as a manuscript in a peer-reviewed scientific journal. 

Reviewer comment: Add registration number in method section. 

Author response: Thank you for this suggestion. However, to our knowledge, Figshare does not provide a registration number for protocols in the same way as other registers, e.g., PROSPERO. Instead, a doi is provided to facilitate citation. This doi/reference has already been cited in the manuscript, on page 8, line 177. The decision to register our protocol with Figshare is in alignment with recommendations within the JBI Manual for Evidence Synthesis, section 11.2. Scoping review protocols are not currently eligible for registration on PROSPERO. 

Reviewer comment: State plan for documenting important protocol amendments. 

Author response: Thank you – added (page 8, lines 178-181). 

Reviewer comment: I think it is better to add the PRISMA 2020 flow diagram for reporting screening result. 

Author response: Thank you – this is indeed how we intend to present our results, see line 317 of the manuscript. Further wording has been added to clarify this/ avoid confusion. 

Reviewer comment: What Quality appraisal tool will be used to evaluate the quality of the eligible studies. 

Author response: A quality appraisal tool will not be used to evaluate the eligible studies, due to the broad, exploratory nature of this review and the large heterogeneity of evidence identified (in terms of study design, methodology and populations – including both quantitative and qualitative sources of evidence). As indicated in the JBI Manual for Evidence Synthesis Appendix 11.2 PRISMA ScR Extension Fillable Checklist, a critical appraisal of individual sources of evidence is not required for a scoping review. Therefore, based on our aims and research questions (and considering our limited resources/time) we believe that a full critical appraisal was not warranted in this instance. However, we would recommend that any follow-up systematic reviews, with more specific research questions regarding effectiveness of telerehabilitation interventions, indeed perform appropriate risk of bias assessment. This decision will be justified in the final manuscript of the results.

---

## [Editor Report · Decision Letter 1]

20 Mar 2024

Rehabilitation interventions delivered via telehealth to support self-management of rheumatic and musculoskeletal diseases: A scoping review protocol

PONE-D-23-09456R1

Dear Dr. Rosemarie Barnett,

We’re pleased to inform you that your manuscript has been judged scientifically suitable for publication and will be formally accepted for publication once it meets all outstanding technical requirements.

Kind regards,

Mehrnaz Kajbafvala, Ph.D

Academic Editor

PLOS ONE
---

## [Editor Report · Acceptance letter]

4 Apr 2024

PONE-D-23-09456R1 

PLOS ONE

Dear Dr. Barnett, 

I'm pleased to inform you that your manuscript has been deemed suitable for publication in PLOS ONE. Congratulations! Your manuscript is now being handed over to our production team.

Kind regards, 

on behalf of

Dr. Mehrnaz Kajbafvala 

Academic Editor

PLOS ONE